# Robust and durable serological response following pediatric SARS-CoV-2 infection

Hanna Renk [1,21], Alex Dulovic [2,21], Alina Seidel[3,21], Matthias Becker [2], Dorit Fabricius[4], Maria Zernickel[4], Daniel Junker[2], Rüdiger Groß [3], Janis Müller [3], Alexander Hilger[5], Sebastian F. N. Bode [4], Linus Fritsch [5], Pauline Frieh[5], Anneke Haddad [5], Tessa Görne[5], Jonathan Remppis[1], Tina Ganzemueller[6], Andrea Dietz [7], Daniela Huzly[8], Hartmut Hengel[8], Klaus Kaier[9], Susanne Weber[9], Eva-Maria Jacobsen[4], Philipp D. Kaiser[2], Bjoern Traenkle[2], Ulrich Rothbauer [2], Maximilian Stich [10], Burkhard Tönshoff[10], Georg F. Hoffmann[10], Barbara Müller[11], Carolin Ludwig[12,13,14], Bernd Jahrsdörfer[12,13,14], Hubert Schrezenmeier[12,13,14], Andreas Peter[15], Sebastian Hörber [15], Thomas Iftner[6], Jan Münch [3], Thomas Stamminger[7], Hans-Jürgen Groß[16], Martin Wolkewitz[9], Corinna Engel[1,17], Weimin Liu[18], Marta Rizzi[19], Beatrice H. Hahn [18], Philipp Henneke [5,20], Axel R. Franz [1,17], Klaus-Michael Debatin[4], Nicole Schneiderhan-Marra [2], Ales Janda[4,22] & Roland Elling [5,20,22✉]

The quality and persistence of children's humoral immune response following SARS-CoV-2 infection remains largely unknown but will be crucial to guide pediatric SARS-CoV-2 vaccination programs. Here, we examine 548 children and 717 adults within 328 households with at least one member with a previous laboratory-confirmed SARS-CoV-2 infection. We assess serological response at 3–4 months and 11–12 months after infection using a bead-based multiplex immunoassay for 23 human coronavirus antigens including SARS-CoV-2 and its Variants of Concern (VOC) and endemic human coronaviruses (HCoVs), and additionally by three commercial SARS-CoV-2 antibody assays. Neutralization against wild type SARS-CoV-2 and the Delta VOC are analysed in a pseudotyped virus assay. Children, compared to adults, are five times more likely to be asymptomatic, and have higher specific antibody levels which persist longer (96.2% versus 82.9% still seropositive 11–12 months post infection). Of note, symptomatic and asymptomatic infections induce similar humoral responses in all age groups. SARS-CoV-2 infection occurs independent of HCoV serostatus. Neutralization responses of children and adults are similar, although neutralization is reduced for both against the Delta VOC. Overall, the long-term humoral immune response to SARS-CoV-2 infection in children is of longer duration than in adults even after asymptomatic infection.

A full list of author affiliations appears at the end of the paper.

To date, our knowledge of children's immune response to infection with severe acute respiratory syndrome coronavirus type 2 (SARS-CoV-2) remains incomplete. In light of current debates on vaccination strategies and non-pharmaceutical preventative measures (e.g. school closures), a comprehensive understanding of protective immunity after natural infection in children is required. As with other viral infections, immune control of SARS-CoV-2 is achieved through a concerted interplay of humoral and cellular immunity[1]. Neutralizing antibodies in children are of particular interest in this context, given their role in blocking virus entry into cells by inhibiting the interaction between the viral receptor binding domain (RBD) within the S-glycoprotein and the angiotensin-converting enzyme 2 (ACE2) receptor[2].

Previous longitudinal studies of the humoral response have found that neutralizing antibodies peak within 3–5 weeks post-infection with a calculated half-life of up to 8 months, suggesting long-term protection in convalescent individuals[1,3–5]. However, most studies only included adults, and longitudinal studies on SARS-CoV-2 infections in children had limited sample size and duration of follow-up post-infection[6–15]. Furthermore, it remains unclear as to whether any form of cross-protection is offered by endemic human coronaviruses (HCoVs) that regularly circulate in the pediatric population, with some studies identifying cross-protection and others not[16,17].

To provide an in-depth characterization of the humoral response in children, we initiated a multi-center longitudinal study, encompassing 328 households each with at least one SARS-CoV-2-infected member, which were followed for up to 12 months after the first infection in each household. This cohort is unique as the subjects exhibited mainly asymptomatic or mild disease with uninfected family members serving as environmental and age-matched controls. We performed an extensive serological evaluation of SARS-CoV-2 infection in all household members, comprising analyses of production of antibodies against various SARS-CoV-2 antigens, including Variants of Concern (VOCs), production of neutralizing antibodies and the role of HCoVs.

## Results

A total of 548 children and 717 adults from 328 households were examined at T1 and 279 households including 402 children and 569 adults were followed to T2 (see Methods and Appendix for full details, Table 1 for a description of the study population, Fig. S2 in the Supplementary Appendix for details on the age structure of the study population). Children were substantially less often seropositive (33.0% at T1, 37.3% at T2) than adults (57.7% at T1, 49.4% at T2) (Table 1). Seropositive participants were almost exclusively mildly or asymptomatically infected. In seropositive individuals, asymptomatic infections were five times more common in children (44.8% T1, 46.0% T2) than in adults (8.7% T1, 11.0% T2) (Table 1), with the proportion of asymptomatic infections decreasing with increasing age (Fig. S3). Overall, hospitalization was rare (3·6% of adults, 0% of children, Table 1). The performance of the four serological assays for children and adults at T1 and T2 is shown in Table S1 and Fig. S4.

The detailed humoral immune response against different SARS-CoV-2 antigens, assessed by MULTICOV-AB is shown in Fig. 1. Children had significantly higher antibody titers against spike ($p < 0.001$), RBD ($p < 0.001$), S1 domain ($p < 0.001$) and nucleocapsid ($p = 0.01$) compared to adults at T1. This increased response was confirmed by the three commercial assays (Fig. S5). In addition, we observed a large difference in seroreversion, with only 3.8% of children, but 17.1% of adults seroreverting between T1 and T2 (Table 1). Seroreversion was not associated with the

response to particular antigens, although the largest and smallest decay in antibody concentrations were observed for antibodies against the S2 domain and nucleocapsid, respectively, regardless of age (Fig. S6).

For both children and adults, there was no significant difference in antibody response between symptomatic and asymptomatic infections (Figs. 2a, b, S7). The frequency of reported symptoms differed between adults and children and the predictive value of each symptom varied between both groups (Fig. 2c, d). While any of the symptoms fever, cough, diarrhea or dysgeusia proved to be a good indicator of infection in adults, dysgeusia was by far the best predictive symptom in children (87.50% of children with dysgeusia were seropositive; 95% CI 71.4–95.2%, 30.5% of children without dysgeusia were seropositive for SARS-CoV-2, 95% CI 29.7–31.3% Fig. 2d). Conversely, cough was a poor predictor of SARS-CoV-2 infection in children (37.4% of children with a cough were seropositive; 95% CI 29.3–46.3%, 33.0% of children without a cough were seropositive; 95% CI 31.0–35.2%, Fig. 2d). Further examination of predictive symptoms among children showed that in contrast to dysgeusia, cough only gained predictive value in children above the age of 12 and the predictive value of fever increased with age (Table S2). There was no difference in the humoral response associated with the presence of particular symptoms in either adults or children (Fig. S8).

To further explore differences in the antibodies produced by children and adults, we analyzed their neutralization potential as well as their binding towards VOCs. The neutralizing potential of a subset of children's sera exceeded that of a subset of adults' at T1 ($p < 0.001$) and T2 ($p = 0.02$) (Fig. 3a). However, this could be attributed to antibody titers, as neutralization in children correlated with the S1-directed antibody response (Spearman's rank 0.86, Fig. 3b). There was no difference in antibody binding responses to the RBD of Alpha and Beta VOCs between adults and children, with an identical binding for the Alpha variant compared to wild-type (Spearman's rank 0.95, Fig. 3c) and a reduction in binding for the Beta variant (Spearman's rank 0.69, Fig. 3d). Neutralization capacity in the pseudotyped virus assay was significantly reduced against the Delta VOC compared to wild-type in both children and adults ($p < 0.01$, Fig. 3e, f). However, neutralization was present in the majority of the seropositive participants–both adults (77.5%) and children (82.0%).

Seroprevalence against endemic coronaviruses rose sharply with age in early childhood, and was stable in older children, adolescents and adults independent of age (Figs. 4a and S9). In contrast to SARS-CoV-2 seroreversion, HCoV antibody titers decreased faster in younger children than in adults (Fig. S10). There were HCoV naïve samples in this cohort and some individuals showed a substantial increase in HCoV antibody response indicating exposure towards endemic HCoVs between the two time points (Fig. 4b in red, Fig. S11). Amongst SARS-CoV-2 exposed individuals in households with a defined index case (index cases excluded from the analysis, see Methods), there was no difference in HCoV antibody titers between SARS-CoV-2 seropositive and seronegative children or adults ($p = 0.21$, Figs. 4c and S12). In addition, we assessed whether SARS-CoV-2 infection boosted HCoV antibody responses, however there was no evidence for an association between HCoV antibody responses and SARS-CoV-2 antibody responses in exposed children or adults (Spearman's rank 0.03, Fig. 4d).

## Discussion

To our knowledge, this is the largest prospective multi-center study comprehensively comparing the adult and pediatric longitudinal humoral immune response following SARS-CoV-2

**Table 1 Demographics and key information for the study cohort.**

|  | Time point T1 | Time point T1 | Time point T2 | Time point T2 |
|---|---|---|---|---|
|  | Adult (717) | Children (548) | Adult (569) | Children (402) |
| Number of participants by age group (n) |  |  |  |  |
| Median Age – years (IQR) | 44 (37–49) | 10 (6–13) | 45 (38–50) | 10 (6–14) |
| Number of females (%) | 362 (50.5) | 277 (50.6) | 297 (52.2) | 202 (50.3) |
| BMI (IQR) | 25.4 (22.2-27.7) | 17.4 (14.9-19.5) | 24.7 (22.3-28.1) | 17.0 (15.0-19.7) |
| Number of seropositive participants | 414 (57.7) | 181 (33.0) | 281 (49.4) | 150 (37.3) |
| • Asymptomatic (%) | 36 (8.7) | 81 (44.8) | 31 (11.0) | 69 (46.0) |
| • Symptomatic (%) | 378 (91.3) | 100 (55.2) | 250 (89.0) | 81 (54.0) |
| Seroreverted at T2 (%) | NA | NA | 71 (17.1) | 7 (3.8) |
| Symptoms at disease onset (of seropositive) | .. | .. | .. | .. |
| • Fever (%) | 217 (52.4) | 66 (36.5) | 151 (53.7) | 49 (32.7) |
| • Cough (%) | 221 (53.4) | 37 (20.4) | 154 (54.8) | 33 (22.0) |
| • Dysgeusia (%) | 266 (64.3) | 28 (15.5) | 176 (62.6) | 24 (16.0) |
| • Diarrhea (%) | 75 (18.1) | 18 (9.9) | 55 (19.6) | 16 (10.7) |
| Median (IQR) days from positive PCR test result to timepoint | 96 (63–120) | 96 (63–120) | 333 (319–353) | 333 (319–353) |
| Median (IQR) days from symptoms onset to timepoint (of seropositive) | 109 (67–122) | 109 (67–122) | 340 (322–356) | 340 (322–356) |
| Hospitalized (of seropositive) (%) | 15 (3.6) | 0 (0.0) | NA | NA |
| Vaccinated (%) | NA | NA | 24 (4.2) | 1 (0.3) |
| Number of households | 328 | 328 | 279 | 279 |
| Median (IQR) number of household members | 4 (3-4) | 4 (3-4) | 4 (3-4) | 4 (3-4) |

See methods for definition of how samples were defined as being seropositive, asymptomatic or symptomatic. Median time from positive PCR test to time point ($n = 368$ at T1, $n = 310$ at T2) and median time from symptoms onset to time point ($n = 349$ at T1, $n = 243$ at T2) are calculated using adult samples for which this data was available. Percentages of seropositive participants refer to the sample size at Time point 1 and Time point 2, respectively.
BMI Body Mass Index, IQR Interquartile Range, NA not applicable, PCR Polymerase Chain Reaction.

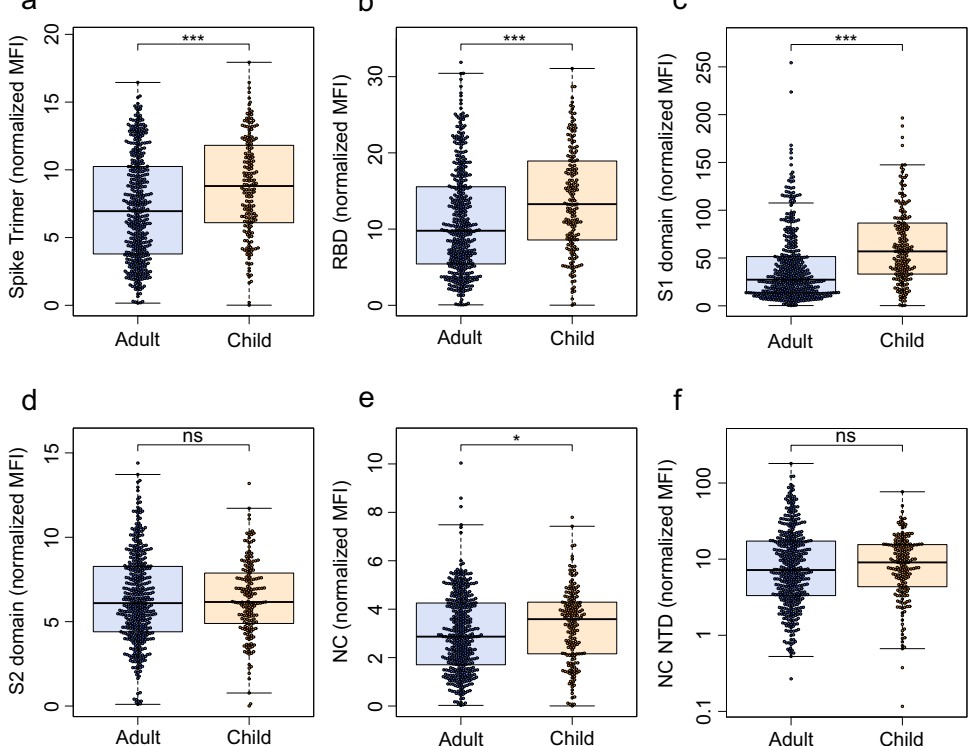

**Fig. 1 Children have a significantly higher humoral response to SARS-CoV-2 than adults.** The humoral response generated following SARS-CoV-2 household exposure with seroconversion was examined using MULTICOV-AB. Children (orange, $n = 181$) produced significantly more antibodies against the Spike (**a** $p = 6.00 \times 10^{-6}$), Receptor Binding Domain (RBD) (**b** $p = 2.86 \times 10^{-6}$), S1 domain (**c** $p = 3.00 \times 10^{-14}$) and nucleocapsid (NC) (**e** $p = 1.76 \times 10^{-2}$) than adults (blue, $n = 414$). There was no significant difference for either the S2 domain (**d** $p = 0.66$) or the N-terminal domain of the nucleocapsid (NC NTD) (**f** $p = 0.40$). Only samples from T1 with a seropositive status (see Methods) are shown. Box and whisker plots with the box representing the median, 25th and 75th percentiles, while whiskers show the largest and smallest non-outlier values. Outliers were identified using upper/lower quartile ±1.5 times IQR. Statistical significance was calculated using Mann–Whitney-U (two-sided) with significance defined as being *<0.05, ***<0.001. Values >0.05 were defined as non-significant (ns). MFI Median Fluorescence Intensity.

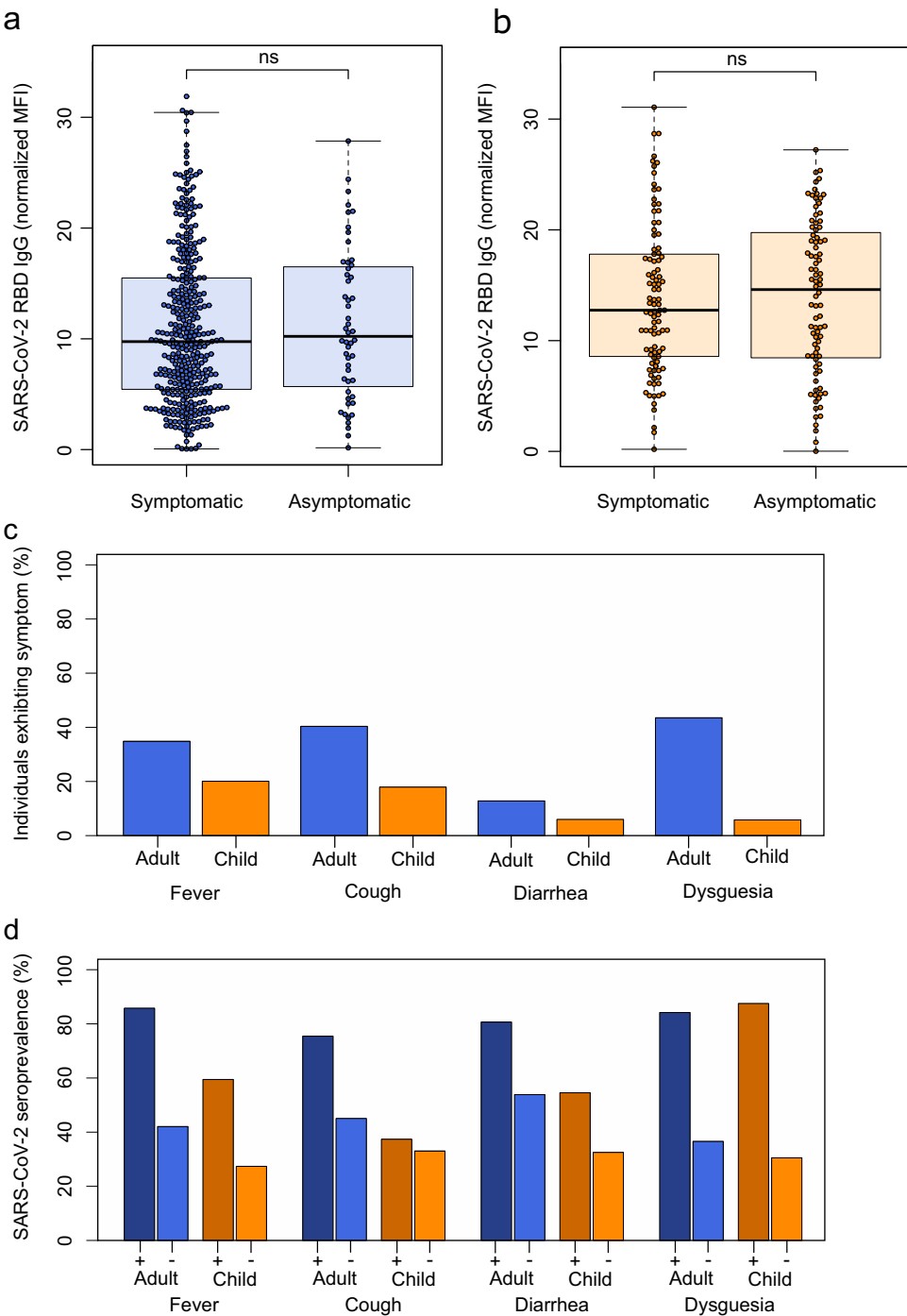

**Fig. 2 SARS-CoV-2 infections in children are more often asymptomatic than in adults, although dysgeusia is a good indicator of SARS-CoV-2 infection in both adults and children.** Box and whisker plots showing that there is no difference in antibody response between asymptomatic and symptomatic SARS-CoV-2 infections in adults (**a** in blue, $p = 0.684$, $n = 414$) or children (**b** in orange, $p = 0.712$, $n = 181$), as assessed by MULTICOV-AB. The receptor binding domain (RBD) is shown as an example, all other SARS-CoV-2 antigens are shown in Fig. S7. Boxes represent the median, 25th and 75th percentiles, while whiskers show the largest and smallest non-outlier values. Outliers were identified using upper/lower quartile ±1.5 times IQR. Statistical significance was calculated using Mann–Whitney-*U* (two-sided). ns indicates a non-significant *p* value >0.05. The four symptoms reported in this study were then examined for their frequency within the study population (**c**), with all symptoms more commonly reported in seropositive adults (in blue) than seropositive children (in orange). Each symptom was then examined for its predictive ability to indicate SARS-CoV-2 infection (**d**), with dysgeusia a strong predictor in both adults (dark blue, 84.2%) and children (dark orange, 87·5%). All other symptoms were poor predictors in children (fever 59.5%, cough 37.4%, diarrhea 54.6%) compared to adults (fever 85.8%, cough 75.0%, diarrhea 80.7%). Only samples from T1 were analyzed for this figure ($n = 717$ adults, 548 children). "+" indicates presence of the symptom "−" indicates absence of the symptom. MFI Median Fluorescence Intensity.

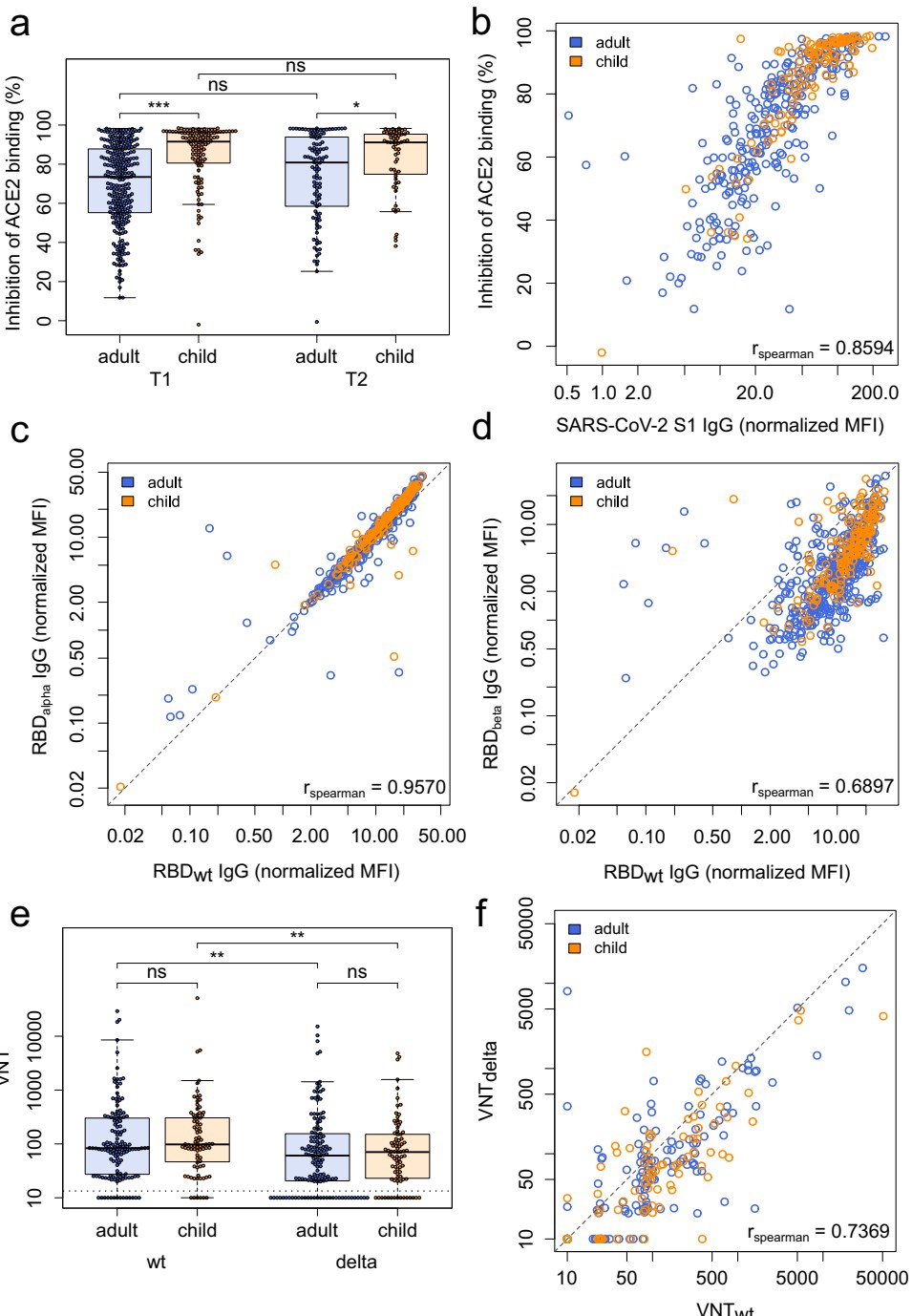

household exposure. As the humoral immunity against SARS-CoV-2 is now increasingly accepted as the central correlate of protection[18–20], improving our incomplete understanding in children[21,22] is of considerable value for public health and vaccination strategies. Importantly, our outpatient cohort has high epidemiological relevance, as a mild course is the most frequent outcome of SARS-CoV-2 infection overall[23]. Our findings identify several unique features of the pediatric serological immune response against SARS-CoV-2.

Children had a lower seroprevalence after household exposure and seropositivity followed asymptomatic infection more frequently than in adults. This is in agreement with our previous report of a different cohort consisting of parent–child pairs[24]. In light of potential pediatric vaccination campaigns, children's

humoral response to SARS-CoV-2 is markedly increased in both quantity and longevity, with children seroreverting at a significantly lower pace than adults. Children generated higher titers of SARS-CoV-2 antibodies than their parents after being exposed to likely same viral strain, and antibody titers negatively correlated with age. Of particular interest are the increase in antibodies produced against the S1 domain and RBD, both of which are associated with higher neutralization capacity, indicating that children produce a high quality humoral response against SARS-CoV-2[2,18,25]. The quality of the pediatric humoral response is further illustrated by the similar binding capacity against the SARS-CoV-2 Alpha and Beta VOCs and a similar neutralization capacity towards the Delta VOC compared to adults. These data argue for the generation of a protective long-term humoral

**Fig. 3 Children and adults produce antibodies with equal neutralizing potential and their antibodies offer the same protection against Variants of Concern. a** Box and whisker plot showing that antibodies produced by children (orange, $n = 118$) have a significantly higher inhibition of ACE2 binding than those produced by adults (blue, $n = 267$, $p = 4.37 \times 10^{-13}$) at T1 and T2 ($p = 0.02$, child $n = 59$, adult $n = 106$) as determined by the sVNT assay. Boxes represent the median, 25th and 75th percentiles, while whiskers show the largest and smallest non-outlier values. Outliers were identified using upper/lower quartile ±1.5 times IQR. Statistical significance was calculated using Mann–Whitney-$U$ (two-sided) with *** indicating a p value < 0.001, * indicating a p value < 0.05, and ns indicating a non-significant p value > 0.05. To determine whether this was due to the higher titers in children, SARS-CoV-2 S1 humoral response was determined using MULTICOV-AB for T1 and plotted against the results of the sVNT assay (**b**). Spearman's rank was calculated to measure the ordinal association between them, confirming that the increase in neutralization is due to higher titers. Protection against the Alpha (**c**) and Beta (**d**) VOCs was determined by MULTICOV-AB and plotted as a linear regression against the antibody binding response to the wild-type (wt) receptor binding domain (RBD), with Spearman's rank calculated to measure the ordinal association. There was no difference in antibody response between children ($n = 166$, T1 samples only) and adults ($n = 381$, T1 samples only) for either variant. (**e**) Box and whisker plot showing reduced neutralization responses in both adults (blue, $n = 142$, $p = 4.38 \times 10^{-3}$) and children (orange, $n = 83$, $p = 6.36 \times 10^{-3}$) against Delta VOC as compared to WT as determined by a pseudotype virus assay (VNT). Boxes represent the median, 25th and 75th percentiles, while whiskers show the largest and smallest non-outlier values. Outliers were identified using upper/lower quartile ±1.5 times IQR. Statistical significance was calculated using Mann–Whitney-$U$ (two-sided) with ** indicating a p value < 0·01 and ns indicating a non-significant value >0.05. Titers are given as serum dilution factor resulting in 50% pseudovirus neutralization (PVNT50). The dashed line represents the lower limit of detection. **f** Linear regression comparing wild-type (VNTwt) and delta (VNTdelta) neutralization responses with Spearman's rank calculated to measure the ordinal association. ACE2 angiotensin-converting enzyme 2, MFI Median Fluorescence Intensity, (s)VNT (surrogate) Virus Neutralization Test, wt wild type.

---

immune response also against VOCs after wild-type SARS-CoV-2 infection, but the quality and duration of this protection can only be estimated following known exposure to VOCs. Children also had significantly higher neutralizing antibody titers than adults, indicating increased protection. This increase in neutralization was directly correlated with higher antibody titers in the other assays, and therefore may not be due to substantial qualitative changes of the pediatric antibody profiles. These findings are in line with one preprinted study[26] but in contrast to two previous studies, which found that children generated a lower humoral response to SARS-CoV-2 than adults, with a corresponding reduction in neutralization activity[10,12]. However, compared to our cohort, all three studies were substantially smaller in sample size and the latter two investigated a different disease spectrum comprising mostly hospitalized children or those diagnosed with hyperinflammatory MIS-C syndrome, and sampled blood at earlier time points after presumed infection.

It is striking that antibody levels in seropositive individuals were independent of fever, cough or diarrhea, as clinical proxies for systemic or localized inflammation of the respiratory or gastrointestinal tract, respectively. Previous studies have reported a clear correlation between disease severity and neutralizing antibody titers in adults[5,27]. At the other end of the disease spectrum with mildly affected younger adults and children as in our cohort, this association was not detectable irrespective of age. This diverges from the classical infection immunology dogma that systemic pathogen–host interaction is required for the generation of robust immune memory. While titers themselves did not differ between asymptomatic and symptomatic infections, we found substantial differences in titers between adults and children. Presence of any symptom was predictive of seropositivity in adults, whereas children showed substantial differences in both the prevalence of symptoms in seropositive individuals and the predictive values of symptoms with respect to SARS-CoV-2 seropositivity. Since cough was a relatively common symptom in children irrespective of seroconversion, it was not useful in predicting SARS-CoV-2 infection. In contrast, dysgeusia, an infrequent symptom among children, was highly accurate in predicting infection. These findings suggest that symptom criteria used for subsequent PCR testing need to be different for children and adults.

Similarly to other authors[28], we identified that exposure to HCoVs, as measured by seropositivity typically happens within the first five years of life. The relatively small decline of HCoV antibody levels during the study period, especially in the adult population, in comparison to the decline in SARS-CoV-2 antibody levels after a single infection suggests that long-term serological immunity against HCoVs may be driven by recurrent exposure. We observed HCoV infections in previously naïve individuals, indicating that endemic HCoVs still circulated between T1 and T2 despite SARS-CoV-2 related distancing and hygiene measures. Although cross-reactivity and/or cross-protection between SARS-CoV-2 and HCoVs have been hypothesized, our analyses did not find evidence for such effects. For both Alpha- and Beta-coronaviruses, HCoV antibody responses were not associated with a lower likelihood of seroconversion following SARS-CoV-2 exposure. Along with frequent HCoV seronegativity in younger childhood, this strongly argues that the lower incidence of SARS-CoV-2 infection in children is not due to HCoV cross-protection. Moreover, there was no evidence for boosting of HCoV titers following SARS-CoV-2 infection. In contrast to other studies which did identify an effect for endemic HCoV infection, our cohort is composed of intensely exposed individuals from within the same households, which is a substantial strength compared to previous studies that have used pre-pandemic sera or indirect control groups[16,26,29].

Limitations of our study include the potential recall-bias inherent to retrospective self- or parent-reporting of symptoms via questionnaires and physician-interviews. Additionally, PCR tests for SARS-CoV-2 during the first wave in Germany were mostly limited to the household index case, meaning it is possible that infected individuals were not identified as such, despite the multi-assay serological approach. However, the in-depth characterization of the humoral response provides valuable data for clinicians, public health officials and the public, at a time when children are increasingly viewed as a potential viral reservoir due to exclusion of pediatric populations from current vaccination strategies. Moreover, the limited PCR capacities make it also possible that non-index study participants were additionally exposed to unknown infection sources outside of their household. Given the relatively low incidence in Germany during the first wave (peak incidence spring 2020 45 cases/100.000/7 days) and the strict lockdown measures including school closures, this scenario can be assumed to be a rare event. Similarly, while PCR testing was not available for all individuals, the strength of this cohort comes from the comparatively large number of children, inclusion of children and adults from the same household, the inclusion of seronegative household members as well-matched controls, and the prospective longitudinal analysis of the humoral response in children for up to one year post-infection. The

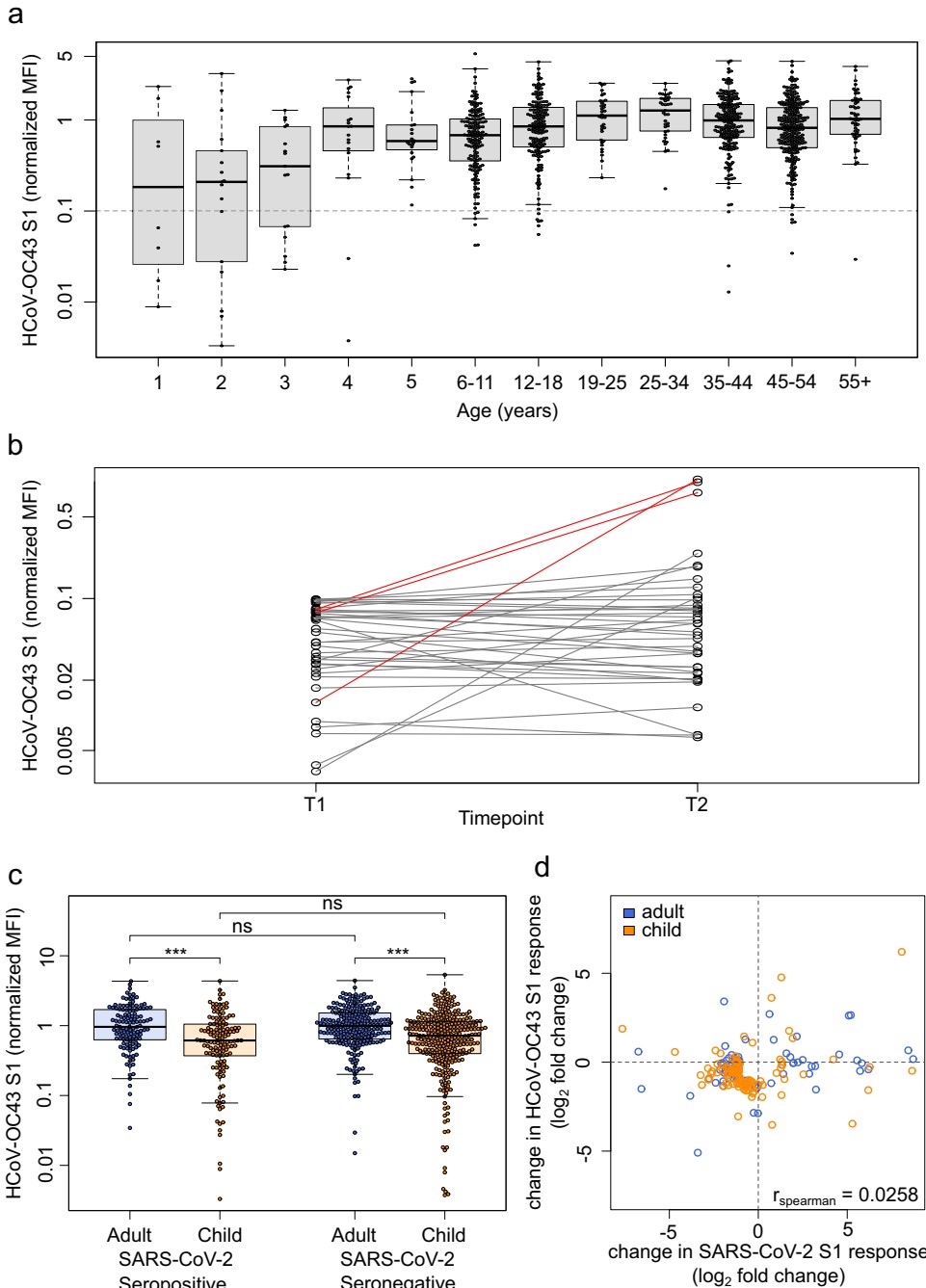

seropositive cohort also comprises almost exclusively individuals with mild or asymptomatic infections and so provides real-world data representative for the majority of SARS-CoV-2 infections in the community. It should be also stated that while all samples were analysed using a range of serological assays, only a subset of samples were analysed for their neutralizing capabilities, and as such, caution should be applied in extrapolating the implications regarding the neutralizing response to all study participants.

In summary, although children mostly show mild or even asymptomatic clinical courses following SARS-CoV-2 infection, they mount a strong and enduring humoral immune response. This strongly argues for sustained protection after infection, and might inform the design of vaccination strategies for SARS-CoV-2 convalescent children.

## Methods

**Cohort**. This study forms part of a non-interventional, prospective observational national multi-center cohort study[30], including 548 children and 717 adults from 328 households each with at least one individual with a SARS-CoV-2 reverse-transcriptase polymerase chain reaction (RT-PCR) proven infection and/or a symptomatic and later serologically proven infection. Participants were recruited during the first wave of the pandemic (May to August 2020) via local health authorities and an in-hospital database of households with at least one laboratory-confirmed SARS-CoV-2 infection. Due to restrictions in obtaining a SARS-CoV-2 RT-PCR test during the first wave (children and asymptomatic contacts of an index case were not tested routinely), serological assays were the only means to identify previous infection. This study was initiated by the four University Children's Hospitals of Freiburg, Heidelberg, Tübingen and Ulm and approved by the independent ethics committees of each center. Sera and data for this substudy were collected at the study sites in Freiburg, Tübingen and Ulm. Participants were asked to fill a questionnaire at time point 1 (May– August 2020) and follow-up time point 2 (February–March 2021).

**Fig. 4 HCoVs offer no protection against SARS-CoV-2, nor do they show a boost-back antibody response following SARS-CoV-2 infection.** Samples from households with a known index case ($n = 971$) were examined with MULTICOV-AB to determine whether the antibody response to endemic coronaviruses (HCoV) provides any protection against infection with SARS-CoV-2. Initial screening of the population showed that seroprevalence increases with age, although several samples were within the blank range of the HCoV assays, indicating the presence of naïve samples (**a**). Naïve samples were defined as those having less than one-tenth the mean antibody response (indicated by dotted line), with the majority of these samples occurring in children under the age of five. HCoV-OC43 is shown as an example, all other HCoVs can be found as Fig. S9. Boxes represent the median, 25th and 75th percentiles, while whiskers show the largest and smallest non-outlier values. Outliers were identified using upper/lower quartile ±1.5 times IQR. **b** Line graph showing the longitudinal response of these naïve samples from T1 to T2, with new infections in HCoV-OC43 shown in red. **c** Box and whisker plot showing there is no significant difference in HCoV-OC43 antibody response between SARS-CoV-2 seropositive and seronegative individuals, among either adults (blue, $n = 440$, $p = 0.974$) or children (orange, $n = 436$, $p = 0.214$). Boxes represent the median, 25th and 75th percentiles, while whiskers show the largest and smallest non-outlier values. Outliers were identified using upper/lower quartile ±1.5 times IQR. Statistical significance was calculated by Mann–Whitney-$U$ (two-sided) with *** indicating a $p$ value $< 0.001$ and ns indicating a $p$ value $> 0.05$. **d** When comparing paired samples longitudinally within the SARS-CoV-2 seropositive subgroup, there was no increase in HCoV-OC43 S1 response in either adults (blue, $n = 76$) or children (orange, $n = 103$) following SARS-CoV-2 infection. Change in response is presented as log2-fold change from T1 to T2 and only samples with either log2-fold change greater than 1 or smaller than −1 are shown. Spearman's rank was used to calculate the ordinal association between the change in response for HCoV-OC43 and SARS-CoV-2. The same figures for the endemic coronaviruses HCoV-NL63, HCoV-HKU1 and HCoV-229E can be found as Figs. S9, S11 and S12. HCoV human Coronavirus, MFI Median Fluorescence Intensity, S1 Spike S1 domain., S1 Spike S1.

**Study participants and eligibility criteria.** Families were identified during the first wave of the pandemic between May and August 2020 in the region of Baden–Württemberg, Germany.

Inclusion criteria:

(i) Children (male or female) aged 1–18 years.
(ii) Parents and other adults (male or female) living in the same household with the investigated children (without age limit).
(iii) Residency in the state of Baden-Württemberg.
(iv) Written consent to the study.

Key exclusion criteria:

(i) Severe congenital diseases (e.g. infantile cerebral palsy, severe congenital malformations).
(ii) Congenital or acquired immunodeficiencies.
(iii) Insufficient comprehension of German language.

**Data collection.** Children and adults within eligible households completed a questionnaire containing demographic information (date of birth, gender, height, weight, smoking), the presence of symptoms (fever, cough, dysgeusia or diarrhea) in plausible temporal association (max. two weeks prior or later) with the onset of the SARS-CoV-2 infection within the household or around the time of a positive SARS-CoV-2 RT-PCR, and symptom duration. For younger children, parents provided symptom information. They additionally provided serum samples for immunological analysis at time point 1 and at follow-up time point 2 after the SARS-CoV-2 infection within the household. Data on vaccination and potential re-infection within the household were collected at time point 2. We investigated all invited households with at least one child to avoid selection bias. Questionnaires were checked for missing or inadequate data and inconsistencies; where possible, these points were clarified retrospectively with the families. To predetermine the sample size, we used a one-factor variance analysis design. Assuming 1.5 children per household participating in the study and three different age ranges, we aimed at a sample size of ≈200 households to reveal small effect sizes of about 0.1 at a significance level of 5% and a test strength of 80%.

**Data variables.** Samples were defined as being "symptomatic" on the basis of having at least one of four (fever, cough, diarrhea and dysgeusia) symptoms, in addition to a positive serology result. "Asymptomatic" samples were defined as those without any of the above symptoms and a positive serology result. For some households ($n = 272$) an "index case" was defined. This is the first household member to test positive for SARS-CoV-2 by RT-PCR. The majority of index cases in this study were adults (249 of 272). No index was defined for households where additional household members tested positive by RT-PCR within 48 hours of the first positive test or where infection was identified by the combination of positive serology and symptoms only. If a household had a defined index case, then all other household members were considered "exposed". "Time post symptom onset" within a household was calculated as the number of days between the first date of symptom onset in any seropositive individual within a household and the sampling date at T1 and T2 respectively.

**Study oversight.** This part of the study was conducted by the University Children's Hospitals in Freiburg, Tübingen and Ulm, Germany. Ethics approval was obtained from the respective Medical Faculties' independent ethics committees (University of Freiburg: 256/20_201553; University of Tübingen: 293/2020BO2;

University of Ulm: 152/20). Written informed consent was obtained from adult participants and from parents or legal guardians on behalf of their children at both sampling time points. Children's preferences on whether or not to provide a blood sample were respected throughout. This study was registered at the German Clinical Trials Register (DRKS), study ID 00021521, conducted according to the Declaration of Helsinki, and designed, analyzed and reported according to the Strengthening the Reporting of Observational Studies in Epidemiology (STROBE) reporting guidelines. The full study protocol can be found at (https://www.drks.de/drks_web/navigate.do?navigationId=trial.HTML&TRIAL_ID=DRKS00021521).

**Blood sample collection.** Samples were collected at two separate time points, an early time point (T1) at a median of 109 days (IQR 67–122 days) after earliest symptom onset in household and a late time point (T2) at 340 days (IQR 322–356 days) post-symptom onset (Table 1, Fig. S1 in Supplementary Appendix). Blood samples were collected by venipuncture from all consenting adults and children within the study. Serum was separated on the same day by centrifugation, aliquoted and frozen at −80 °C until used.

**Serological assays.** Antibodies against SARS-CoV-2 in 2236 samples were detected using the following four assays: (1) EuroImmun-Anti-SARS-CoV-2 ELISA IgG (S1), (2) Siemens Healthineers SARS-CoV-2 IgG (RBD), (3) Roche Elecsys Ig (Nucleocapsid Pan Ig) and (4) MULTICOV-AB, a previously published multiplex immunoassay that simultaneously analyses antibody binding to 23 antigens from SARS-CoV-2 (including VOCs)[31,32]. Seropositivity was defined as any three of the four SARS-CoV-2 assays being positive. The MULTICOV-AB assay also analyses antibody binding to endemic coronavirus antigens (i.e. HCoV-OC43, -NL63, -HKU1 and −229E)[31,32].

*EuroImmun Anti-SARS-CoV-2 ELISA.* The EuroImmun Anti-SARS-CoV-2 ELISA (IgG) was performed as the manufacturer's instructions to detect IgG antibodies against the S1 domain of the SARS-CoV-2 spike protein. All 2236 samples used in the final analysis were measured with this assay. All samples were processed with the specified controls and calibrators. Serological analysis was performed blinded for all clinical covariables.

*Siemens Healthineers SARS-CoV-2 IgG (sCOVG).* The Siemens sCOVG assay was performed as per the manufacturer's instructions on an Advia Centaur XPT platform to detect IgG antibodies against the receptor-binding-domain (RBD) of the SARS-CoV-2 spike protein. All 2,236 samples used in the final analysis were measured with this assay. All samples were processed with the specified controls and calibrators. Serological analysis was performed blinded for all clinical covariables.

*Roche Elecsys Electrochemiluminescence immunoassay (ECLIA).* The Roche Elecsys ECLIA was performed as per the manufacturer's instruction on a Cobas e411 or e811 platform to detect IgG, IgA and IgM antibodies against the nucleocapsid of SARS-CoV-2. All 2236 samples used in the final analysis were measured with this assay. All samples were processed with the specified controls and calibrators. Serological analysis was performed blinded for all clinical covariables.

*MULTICOV-AB™.* All 2236 samples used in the final analysis were analyzed using MULTICOV-AB™[1], a bead-based multiplex immunoassay that simultaneously analyzes 23 antigens from SARS-CoV-2 (including RBDs from variants of concern and endemic human coronaviruses)[31]. A full list of antigens used in this study can

be found in Table S3. Samples were measured in 384-well plates, with all pipetting steps performed using a Beckmann Coulter i7 pipetting robot. Antigens were coupled by EDC/s-NHS or Anteo coupling to spectrally distinct populations of MagPlex beads (Luminex Technology). Samples were diluted in assay buffer (1:4 Low Cross Buffer (Candor Bioscience GmBH) in CBS (1x PBS + 1% BSA) + 0.05% Tween20) and added to bead mix to a final dilution factor of 1:400, before being incubated for 2 hours at 21 °C on a thermomixer (1500rpm). Unbound antibodies were then removed by washing with Wash buffer (1x PBS, 0.05% Tween20). Bound antibodies were detected using RPE-conjugated human IgG (3 µg/mL) and IgA (5 µg/mL) (both Biozol) by incubation for 45 mins at 21 °C, 1800 rpm on a thermomixer. Following a further washing step, beads were resuspended in 80 µL of washing buffer and shaken briefly for 3 mins at 1500 rpm. Plates were then measured using a FLEXMAP-3D (Luminex Technology) instrument running xPONENT Software (version 4.3) with the following settings: 60 µL, 80 s timeout, 35 events, Gate 7500-15000 and Reporter Gain: Standard PMT. For quality control, eight wells for each QC sample plus eight blank wells (negative control) were included on each 384-well plate[31–33]. Additionally, control beads coupled with human IgG, goat-anti-human IgG, human IgA and goat-anti-human IgA were included in each well to act as controls for both sample addition and signal system addition. To pass QC, each sample had to meet the minimum threshold for number of beads per ID (35), have a sample and signal system control bead value within normal range and pass plate-by-plate QC sample controls. Any plate or sample that failed QC was re-measured (83/2,390). Normalization values for each antigen were generated by dividing the raw median fluorescence intensity (MFI) value by the mean plate-by-plate MFI of QC2 (IgG) or QC3 (IgA). For SARS-CoV-2, normalization values >1 for the trimeric spike and wild-type RBD indicate positivity. To reduce analytical variations, all samples were analyzed in the same run. Serological analysis was performed blinded for all clinical covariables. Technical questions regarding the MULTICOV-AB™assay should be directed to nicole.schneiderhan@nmi.de.

*Surrogate SARS-CoV-2 Neutralization Test*. A subset of 385 samples were analyzed for neutralization with the surrogate SARS-CoV-2 neutralization test (GenScript) as per the manufacturer's instructions and as published previously[34]. Briefly, samples and controls were incubated with an HRP-conjugated RBD fragment. Following this, the mixture was added to wells of a capture plate coated with human ACE2 protein. The plate was then washed three times to remove any complexes or non-bound antibodies. TMB was added and then stopped with the addition of a stop reagent. The plate was then read by a microtiter plate reader (POLARstar Omega) at 450 nm. The absorbance of the sample is inversely correlated with the amount of SARS-CoV-2 neutralizing antibodies. Positive and negative controls served as internal assay quality controls. The test was considered valid only if the OD450 for each control fell within the respective range (OD450negative control >1.0, OD450positive control <0.3). For final interpretation, inhibition rates were calculated as follows: Inhibition score (%) = (1 − (OD valuesample/OD valuenegative control) × 100%). Scores <30% were considered negative, scores ≥30% were considered positive.

**Cell culture**. Vero E6 (African green monkey, female, kidney; CRL-1586, ATCC, RRID:CVCL_0574) cells were grown in Dulbecco's modified Eagle's medium (DMEM, Gibco) supplemented with 2.5% heat-inactivated fetal calf serum (FCS), 100 units/ml penicillin, 100 µg/ml streptomycin, 2 mM L-glutamine, 1 mM sodium pyruvate, and 1x non-essential amino acids. HEK293T (human, female, kidney; ACC-635, DSMZ, RRID: CVCL_0063) cells were grown in DMEM supplemented with 10% FCS, 100 units/ml penicillin, 100 µg/ml streptomycin, and 2 mM L-glutamine. All cells were grown at 37 °C in a 5% CO2 humidified incubator.

**Preparation of pseudotyped particles**. Rhabdoviral pseudotype particles were prepared as previously described[34]. A replication-deficient VSV vector in which the genetic information for VSV-G is replaced by genes encoding enhanced green fluorescent protein and firefly luciferase 3 (kindly provided by Gert Zimmer, Institute of Virology and Immunology, Mittelhäusern, Switzerland) was used for pseudotyping. HEK293T cells were transfected with expression plasmids encoding SARS-CoV-2 spike variants D614G 4 (pCG1_SARS-2-Sdel18_D614G, kindly provided by Stefan Pöhlmann) or B.1.617.2/Delta[35], containing the spike mutations T19R, G142D, E156-, F157-, R158G, L452R, T478K, D614G, P681R, D950N and a 19AA C-terminal deletion (pcDNA3.1-S2S-IN2(B.1.617.2)Δ19. 24 h post transfection, cells were inoculated with VSV vector. After 2 h incubation at 37 °C, the inoculum was removed, cells were washed with PBS and fresh medium added. After 16–18 h, the supernatant was collected and centrifuged (2,000 × g, 5 min, room temperature) to remove cellular debris. Cell culture medium containing anti-VSV-G antibody (I1-hybridoma cells; ATCC no. CRL-2700) was added to block residual VSV-G-containing particles. Samples were then aliquoted and stored at −80 °C.

**Pseudovirus neutralization assay**. A subset of 225 samples were examined by Pseudovirus neutralization assay against Wild-type (B1 isolate) and the delta VOC (B.1.617.2). For pseudovirus neutralization experiments, Vero E6 cells were seeded in 96-well plates one day prior. Heat-inactivated (56 °C, 30 min) sera were serially diluted

in PBS, mixed with pseudovirus stocks (1:1, v/v) and incubated for 30 min at 37 °C before being added to cells. After 16-18 h, firefly luciferase activity was quantified as a readout for transduction efficiency. For this, cells were lysed by incubation with Cell Culture Lysis Reagent (Promega) at room temperature. Lysates were then transferred into white 96-well plates and luciferase activity was measured using a commercially available substrate (Luciferase Assay System, Promega) and a plate luminometer (Orion II Microplate Luminometer running Simplicity Software v4.2, Berthold). For analysis, background signal of untreated cells was subtracted and values normalized to pseudovirus mixed with PBS only. Results are given as serum dilution resulting in 50% pseudovirus neutralization (PVNT50) on cells, calculated by nonlinear regression ([Inhibitor] vs. normalized response–Variable slope) in GraphPad Prism Version 9.1.1. The upper and lower cutoff values of this assay were set at PVNT50 >81,920 and PVNT50 <20, respectively.

**Data analysis**. Initial data collection was done using Microsoft Excel and Access. Formal data analysis was performed on RStudio (Version 1.2.5001, running R 3.6.1) with the following additional packages: "RColorBrewer", "beeswarm", "gplots", "VennDiagram", all of which were used solely for data depiction and not statistical analysis. Figures were generated in RStudio and then edited for clarity in Inkscape (Inkscape 0.92.4). Only samples for which full data for MULTICOV-AB was available were included in the analysis. Furthermore, only samples from time point 1 were used for all non-longitudinal analyses. For longitudinal analyses, only those participants for whom both T1 and T2 samples were available were included and all participants who were vaccinated prior to T2 were excluded. For analysis of potential cross-protection though endemic coronaviruses, only households with a known index case were used and the index case itself was excluded. Statistical analyses performed are described in the figure legends. For comparison of signal distribution between sample groups, Mann–Whitney-U tests were performed using the "wilcox.test" function from R's "stats" library. For correlation analysis, Spearman's rank was calculated using the "cor" function from R's "stats" library. $p$ values < 0.05 were considered to be significant.

**Reporting summary**. Further information on research design is available in the Nature Research Reporting Summary linked to this article.

## Data availability

A short version of the study protocol is available at the German Clinical Trials Register (DRKS, www.drks.de), study ID 00021521. The full study protocol is available from https://www.drks.de/drks_web/navigate.do?navigationId=trial.HTML&TRIAL_ID=DRKS00021521. Individual participant data, including data dictionaries will not be available, since we did not seek parental consent for data sharing.

## Code availability

Code will be available upon reasonable request.

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

## Acknowledgements
We thank Carmen Blum, Sevil Essig, Ulrike Formentini, Jens Gruber, Andrea Hänsler, Simone Hock, Ann Kathrin Horlacher, Jennifer Juengling, Gudrun Kirsch, Ingrid Knape, Helgard Knauss, Sonja Landthaler, Alexandra Niedermeyer, Bianca Rippberger, Andrea Schuster, Boram Song, Ulrike Tengler, Mareike Walenta and Linda Wolf for assistance with sample processing and patient material storage. We are grateful for the FREEZE and HILDA biobank Freiburg for sample processing, in particular Ali-Riza Kaya, Marco Teller and Dirk Lebrecht. We thank Sandra Steinmann, Yvonne Müller, Vanessa Missel at the University Hospital Ulm, Andrea Evers-Bischoff, Andrea Bevot and the CPCS at the University Hospital Tübingen for organizational support in conducting the study. We thank Steffen Keul for assistance with data processing. This work was financially supported by the State Ministry of Baden-Württemberg for Economic Affairs, Labor and Housing Construction (grant numbers FKZ-3-4332.62-NMI-67 and FKZ-3-4332.62-NMI-68) to NSM, the Ministry of Science, Research and the Arts Baden-Württemberg within the framework of the special funding line for COVID-19 research to the Freiburg, Tübingen, Ulm and Heidelberg centers, the Federal Ministry of Health to the Freiburg study site (PH and RE) and the NIH (R01 AI 050529 and R37 AI 150590 to B.H.H). Additional funding by the German Federal Ministry of Education and Research (BMBF 01GL1746B) to PH. The funders had no role in study design, data collection, data analysis or the decision to publish. The funding agencies of the study, the Ministry of Science of the State of Baden-Württemberg, the NIH, and the Ministry for Economic Affairs, Labor and Housing Construction of the State of Baden-Württemberg had no role in study design, data collection, data analysis, interpretation of data, writing of the report, and in the decision to submit the paper for publication.

## Author contributions
R.E., H.R., A.J., D.F., P.H., A.R.F. and K.M.D. conceived the study. H.R., A.D., R.E., A.J., P.H., A.R.F., K.M.D. and N.S.M. designed the experiments. R.E., H.R., A.J., B.H.H., P.H., A.R.F., K.M.D. and N.S.M. procured funding. A.D., M.B., D.J., A.S., R.G., J.M., J.M., A.H., C.L., T.G., A.D., D.H., H.H., A.P., S.H., T.I., T.S., W.L. and H.-J.G. performed experiments. R.E., H.R., A.J., D.F., M.Z., S.B., L.F., P.F., A.H., J.R., E.-M.J., C.E., M.W., T.G. and M.R. collected samples or organized their collection. B.J., H.S., M.S., B.T., G.F.H. and B.M. supported the sample collection and provided key resources. P.K., B.T. and U.R. produced the R.B.D. mutants. H.R., A.J., A.D., M.B., D.F., A.H., A.D., K.K., S.W., E.-M.J., A.P., T.I., T.S., H.-J.G., M.W., C.E., K.M.D. and M.R. curated the data. M.B. and A.D. performed the data analysis. A.D. and M.B. generated the figures. A.D., H.R., A.J. and R.E. wrote the first draft of the manuscript. All authors approved the final version of the manuscript. All authors confirm that they had full access to all the data in the study and accept responsibility to submit for publication.

## Funding

## Competing interests
N.S.M. was a speaker at Luminex user meetings in the past. The Natural and Medical Sciences Institute at the University of Tübingen is involved in applied research projects as a fee for services with Luminex. The other authors report no competing interests.

## Additional information

[1]University Children's Hospital Tübingen, Tübingen, Germany. [2]NMI Natural and Medical Sciences Institute at the University of Tübingen, Reutlingen, Germany. [3]Institute of Molecular Virology, Ulm University Medical Center, Ulm, Germany. [4]Department of Pediatrics and Adolescent Medicine, Ulm University Medical Center, Ulm University, Ulm, Germany. [5]Center for Pediatrics and Adolescent Medicine, Medical Center Freiburg, Germany and Faculty of Medicine, University of Freiburg, Freiburg, Germany. [6]Institute for Medical Virology and Epidemiology of Viral Diseases, University Hospital Tübingen, Tübingen, Germany. [7]Institute of Virology, Ulm University Medical Center, Ulm, Germany. [8]Institute of Virology, Medical Center Freiburg, Germany and Faculty of Medicine, University of Freiburg, Freiburg, Germany. [9]Institute of Medical Biometry and Statistics, Medical Center Freiburg, Germany and Faculty of Medicine, University of Freiburg, Freiburg, Germany. [10]Department of Pediatrics I, University Children's Hospital Heidelberg, Heidelberg, Germany. [11]Department of Infectious Diseases, Virology, Heidelberg University Hospital, Heidelberg, Germany. [12]Institute of Transfusion Medicine, Ulm University, Ulm, Germany. [13]Institute for Clinical Transfusion Medicine and Immunogenetics, Ulm, Germany. [14]German Red Cross Blood Transfusion Service, Baden-Württemberg-Hessen, Germany. [15]Institute for Clinical Chemistry and Pathobiochemistry, University Hospital Tübingen, Tübingen, Germany. [16]Institute of Clinical Chemistry, Ulm University, Ulm, Germany. [17]Center for Pediatric Clinical Studies, University Hospital Tübingen, Tübingen, Germany. [18]Department of Microbiology and Department of Medicine, University of Pennsylvania, Philadelphia, USA. [19]Department of Rheumatology and Clinical Immunology, Medical Center Freiburg, Germany and Faculty of Medicine, University of Freiburg, Freiburg, Germany. [20]Institute for Immunodeficiency, Medical Center Freiburg, Germany and Faculty of Medicine, University of Freiburg, Freiburg, Germany. [21]These authors contributed equally: Hanna Renk, Alex Dulovic, Alina Seidel. [22]These authors jointly supervised this work: Ales Janda, Roland Elling. ✉email: roland.elling@uniklinik-freiburg.de

