## [Peer Review File · Nature Communications]

Robust and durable serological response following pediatric SARS-CoV-2 infectionReviewers' Comments:

Reviewer #1:

Remarks to the Author:

Prospective multi-centre study of community acquired (household) SARS-CoV-2 infection shows that asymptomatic infection was more common in children than in adults. Children had higher SARS-CoV-2 antibody levels which persisted longer compared to adults. Authors found no association between humoral responses to common cold coronaviruses and SARS-CoV-2 arguing against significant cross protection. Similar to previous studies neutralising antibodies to wild type and alpha variants were equivalent however titres were lower against delta variant. Findings are clearly presented and easy to follow. Authors should acknowledge limitation of study as outlined and some of the conclusions from data regarding immune protection are premature and not supported by data presented in this study

Study strengths

Prospective longitudinal study of 328 households with least one SARS-CoV-2 infected member. Findings more representative of community acquired SARS-CoV-2 infection (asymptomatic or mild infection) in contrast to hospital based infection with confounding occupational exposure and symptomatic infection requiring either A&E attendance or admission. In addition use of un-infected family members served as environmental and age matched controls

There are a number of minor limitations which authors should acknowledge

1. Lack of use of WHO international standard for SARS-CoV-2 immunoglobulin for both serology and neutralisation assays which limits reproducibility of findings.

2. Serological assay use semi-quantitative readout. Roche and Siemens commercial ELISAs for example now have quantitative readouts.

3. IgA antibody levels decline more rapidly than IgG: Euroimmune ELISA detects both IgG and IgA and isotype for Roche not stated

4. Suppl Table S1

No of seropositive and seronegative for MULTICOV S alone should be reported

Did all assays used meet national or WHO standards for detection of SARS-CoV-2 antibodies? Was there a significant difference in SARS-CoV-2 sero-prevalence using combination Roche and MULTICOV S as opposed to three out of four assays being positive for SARS-CoV-2 antibodies?

5. What are the performance characteristics of neutralisation assay used compared to plaque reduction neutralisation test (PRNT)?

6. Were there any demographic or immunological characteristics between those who were studied 11–12 months post self-reported infection and those lost to study follow up?

7. Results Line 227-230:

‘However neutralisation was present in majority of seropositive participants (77.5% of seropositive adults and 82.0% children) indicating the generation of protective long term humoral immune responses against these and potentially future VOCs’.

Conclusion should be modified as we do not know how many individuals have been protected against delta variant for example.

8. Discussion Line 261-262

‘Children generated a higher titre of SARS-CoV-2 antibodies after being exposed to same strain and antibody titres are correlated with age.

Sentence could imply that SARS-CoV-2 genotype was performed during the study: children generated higher antibodies to SARS-CoV-2 than index case within household

9. Discussion Lines 265-269

The quality of the paediatric humoral immune response is further illustrated by similar binding capacity against SARS-CoV-1 Alpha and beta VOC and a similar neutralization capacity toward Delta VOC compared to adults. This illustrates the generation of a protective long-term humoral immune response also against VOCs after wild type SARS-CoV-2 infection. Difficult to assess unless children have been exposed to VOCs

Reviewer #2:

Remarks to the Author:

This is a very well written manuscript that discusses the long term persistence of SARS-CoV-2 specific antibodies and correlation with symptoms. The authors compare pediatric and adult immune responses to SARS-CoV-2 and human CoV. The authors conclude that children had comparable immune response to adults, even with asymptomatic infection. This is an important study as it might help with providing guidance around vaccination.

Major comments:

- patients were not regularly tested by PCR so it is possible that patients were infected from sources outside of their household and at different times during the study (separate from the index case). With that in mind it would be helpful if the authors stated whether or not the children and adults were at school or work during the study period.
- an additional limitation that should be mentioned is that children may be less likely to accurately report symptoms than adults

Minor comments:

- Abstract background: suggest clarifying that "long-term persistence of antibodies.... 'are one part' of the immune system that determines..." to clarify that cellular immunity likely also plays an important role
- Introduction: line 88: the authors have a minor error with reference 1
- Results:
 - it would be interesting to know who was the index case in the household (adult versus child) if known and include in results
 - Figure 4: is anything known about the outliers in 4b? Were these individuals more likely to be symptomatic?

Discussion:

- minor typo on line 251 should read as "higher titer"
- it is interesting that fever was not a good predictor of SARS-CoV-2 infection in children as respiratory viruses decreased globally during the pandemic due to mitigation measures. It would be interesting for the authors to comment on why they think this might be.

-Table 1:

- was any information collected on co-morbidities for the study participants?
would include # adults and children per household

Reviewer #3:

Remarks to the Author:

This is a well written, strong manuscript containing a timely analysis, in a topic of interest, especially as immunizations are now due to roll out to younger children. The authors report their investigation of 328 households in Germany, where one member of the household had been identified as being SARS CoV-2 positive, with an 11 to 12-months of serologic follow-up (Serum specimens collected roughly 3 months to 11 months after infection of the index case). Uninfected family members served as age matched and environmental controls although did not see much information about controls in the main text, but there are figures with data on seronegative individuals in the supplement. Study cohort was comprised of 548 children and 717 adults from 328 households each with at least one individual with a

SARS-CoV-2 RT-PCR proven infection and/or a symptomatic and later serologically proven infection. Participants were recruited during the first wave of the pandemic (May to August 2020) via local health authorities and an in-hospital database of households with at least one laboratory-confirmed SARS-CoV-2 infection. Per inclusion criteria it appears that only households with children/ youth between ages 1-18 years were eligible and patients with immunodeficiencies were excluded. This is stated in the methods but figure S2 shows children under age 1 year enrolled so should clarify to be consistent if infants under 1 year of age were enrolled, and if so how many.

The study was conducted during the first wave when no immunizations were available to the population. This would correspond to the time of acute infection. However, immunizations may have been given before collection of specimens in T2 although per Table 1 that was a relatively negligible number of participants (4.2% of adults and 0.3% of children). Only assays measuring antibodies to the nucleocapsid protein would be able to distinguish antibodies from vaccine to antibodies generated by natural infection. Nucleocapsid antibodies would not be detected by the EuroImmuno Anti-SARS-CoV-2 ELISA and the Siemens Healthineers SARS-CoV-2 IgG (sCOVG) which measure antibodies to spike protein. The authors mention that in T2 the assay used for all specimens was the Roche Elecsys Electrochemiluminescence immunoassay (ECLIA) which measures antibodies to the nucleocapsid of SARS-CoV-2 and the MULTICOV-AB™1, a bead-based 136 multiplex immunoassay that simultaneously analyzes 23 antigens from SARS-CoV-2. This information is included in the supplement but should be highlighted in the main manuscript too as it is an important point to make regarding interpretation of results.

Pseudovirus neutralization assays were performed in a subset of 385 specimens, so it is important to include in the study limitation section in the discussion that the neutralization results do not apply to all evaluable serum specimens. The statements in the discussion about neutralization results are very broad and there is no mention there that only a subset of specimens were analyzed.

Overall the results are very interesting and substantiated by other reports in which children were less frequently seropositive (33% and 37% respectively at the 2 timepoints) as compared to adults (58% and 40% respectively at the 2 timepoints). Most participants were asymptomatic or mildly symptomatic with a very low hospitalization rate. Symptoms were associated with increasing age. It is important to remember that this was a cohort from early on in the pandemic, before variants of concern such as Delta strains emerged. Children sustained longer duration of antibody responses than adults in this study, with seroreversion being more frequently seen in adults than children as stated in the results section: "only 3.8% of children, but 17.1% of adults seroreverted between T1 and T2 (Table 1)." However, figure S6 in the supplement section states that seroreversion occurs at the same rate in adults and children, so this is confusing to the reader. The authors should clarify if there are any discrepancies in the text.

Also interesting results was that there was no difference in antibody responses between asymptomatic and symptomatic infections in this cohort. This could be due to the fact that the cohort did not have many participants with severe illness, so the absence of a notable difference could be due to the fact that people were not severely ill in general, and not because symptomatic individuals produce the same amount of antibody that asymptomatic individuals. This should be mentioned in the discussion, otherwise interpretation of results could be misleading. Many studies have shown that individuals with severe illness tend to develop higher titer antibodies over time which the authors do mention in the discussion.

Interesting finding that there was no association between antibody responses to HCoV and SARS-CoV-2 antibody responses in children and adults, which indicates that there is no cross-protection conferred by antibodies to other coronaviruses. This had been postulated as a reason why children are not generally so ill when they acquire SARS-CoV-2. This is an important finding in this study.

There have been other household transmission studies evaluating pediatric and adult populations with

SARS CoV-2 infection, so not clear whether the authors' claim that this is the largest multi-centric study conducted to date in pediatric populations still holds. Some of the other studies were from the early phase of the pandemic such as that of Lugon P et al. SARS-CoV-2 Infection Dynamics in Children and Household Contacts in a Slum in Rio de Janeiro. Pediatrics. 2021. Others were from later pandemic stages with circulation of more contagious VOC such as Liu P et al Pediatric Household Transmission of SARS CoV-2 Infection- Los Angeles County, December 2020 to February 2021. Pediatric Infectious Diseases Journal, October 2021; Dawood FS et al. Incidence rates, household infection risk, and clinical characteristics of SARS CoV-2 infection among children and adults in Utah an New York City, NY. JAMA Pediatrics 2021, among others. It would be of interest if the authors included a paragraph in their discussion comparing their findings with that of the similar cohort studies, such as the ones mentioned above.

Some minor comments:

Introduction, line 88: Would just end sentence with reference rather than state reviewed in 1.

Discussion, line 260: Would rewrite to: with children seroreverting at a significantly slower pace than adults.

Discussion, line 261: Would change to: Children generated higher titers...

Discussion, line 262: Would change to: antibody titers negatively correlated with age.

Point by point reply Renk et al., NCOMMS-21-33874-T

Reviewer #1 (Remarks to the Author):

Prospective multi-centre study of community acquired (household) SARS-CoV-2 infection shows that asymptomatic infection was more common in children than in adults. Children had higher SARS-CoV-2 antibody levels which persisted longer compared to adults. Authors found no association between humoral responses to common cold coronaviruses and SARS-CoV-2 arguing against significant cross protection. Similar to previous studies neutralising antibodies to wild type and alpha variants were equivalent however titres were lower against delta variant. Findings are clearly presented and easy to follow. Authors should acknowledge limitation of study as outlined and some of conclusions from data regarding immune protection are premature and not supported by data presented in this study

We thank the reviewer for their kind words and hope we addressed their minor concerns suitably.

Study strengths

Prospective longitudinal study of 328 households with least one SARS-CoV-2 infected member. Findings more representative of community acquired SARS-CoV-2 infection (asymptomatic or mild infection) in contrast to hospital based infection with confounding occupational exposure and symptomatic infection requiring either A&E attendance or admission. In addition use of un-infected family members served as environmental and age matched controls

There are a number of minor limitations which authors should acknowledge

1. Lack of use of WHO international standard for SARS-CoV-2 immunoglobulin for both serology and neutralisation assays which limits reproducibility of findings.

We completely agree with the reviewer regarding the lack of WHO standards within the assays which could limit reproducibility. For all commercially used assays, we were only able to use what was available from the manufacturer at the time of measurements. For MULTICOV-AB, the WHO standard was only recently incorporated and unfortunately, back-calculation of the AU is not possible due to the lack of required QC sample.

To counteract this, as well as the variable performance of the different assays, the criteria of 3 out of 4 being positive was used to ensure that findings were based upon consistency among multiple assays, and hence were reproducible.

For pseudovirus neutralization assays, all used pseudoparticle stocks were internally validated using the anti-SARS-CoV-2 Spike monoclonal antibody Imdevimab to ensure comparable performance. The Surrogate neutralization assay was again what was available from the manufacturer at the time.

2. Serological assay use semi-quantitative readout. Roche and Siemens commercial ELISA's for example now have quantitative readouts.

The semi-quantitative readout was what was available when this study was designed and the measurements were performed.

3. IgA antibody levels decline more rapidly than IgG: Euroimmune ELISA detects both IgG and IgA and isotype for Roche for not stated

Two EuroImmuno assays were used, one which detects IgG and one which detects IgA, we have written this more clearly in the manuscript. The Roche assay identifies panIg as stated in the methods.

4. Suppl Table S1

No of seropositive and seronegative for MULTICOV S alone should be reported

The MULTICOV-AB assay uses a dual cut-off based upon both the S and RBD antigens to determine seropositivity. Using just the S antigen will not reflect the assay sens/spec correctly. We will therefore not include this within the table.

Did all assay used meet national or WHO standards for detection of SARS-CoV-2 antibodies? Was there a significant difference in SARS-CoV-2 sero-prevalence using combination Roche and MULTICOV S as opposed to three out of 4 assay being positive for SARS-CoV-2 antibodies?

For the commercially available assays, their performance has been verified by various national bodies for use in antibody testing. For MULTICOV-AB, it has been shown to far exceed the WHO standards for detection of antibodies with 90% sensitivity (80% required) and 100% specificity (97% required). The assay was also validated to FDA guidelines. Further information on MULTICOV assay validation and performance can be found in Becker et al, Nature Comms 2021.

As expected, more samples are classified as positive if only MULTICOV and Roche are used due to their superior sensitivity and specificity, however this is non-significant, with 1080 (48.3%) samples classified as positive as opposed to 1033 (46.2%) when the 3 of 4 method is used.

5. What are the performance characteristic of neutralisation assay used compared to plaque reduction neutralisation test (PRNT)?

This question has already been addressed by several studies (Riepler et al. 2020, doi: 10.3390/vaccines9010013) (Schmidt et al. 2020, doi: 10.1084/jem.20201181). Neutralization assays using VSV-based particles pseudotyped with SARS-CoV-2 spike, such as the one performed in our study, have been shown to yield robust and reproducible results. In addition, the obtained titers have been proven to strongly correlate with the results from neutralization assays using authentic SARS-CoV-2, including a focus forming unit (FFU) assay.

6. Were there any demographic or immunological characteristics between who were studies 11—12 months post self-reported infection and those lost to study follow up?

There were no significant demographic or immunological characteristics between those who were followed up and those were not.

7. Results Line 227-230:

'However neutralisation was present in majority of seropositive participants (77.5% of seropositive adults and 82.0% children) indicating the generation of protective long term humoral immune responses against these and potentially future VOCs'.

Conclusion should be modified as we do not know how many individuals have bene protected against delta variant for example.

We agree with the Reviewer that we do not have direct evidence for this conclusion and have therefore omitted it.

8. Discussion Line 261-262

'Children generated a higher titre of SARS-CoV-2 antibodies after being exposed to same strain and antibody titres are correlated with age.

Sentence could imply that SARS-CoV-2 genotype was performed during the study: children generated higher antibodies to SARS-CoV2 than index case within household

Given the relatively low incidence of SARS-CoV-2 infections in Germany at the initial sampling timepoint, as well as the strict contact reduction measures that were in place (e.g. school closures), it can be assumed that all household members were exposed to the same viral strain. However, we agree with the reviewer that as sequencing was not performed, and we therefore could not formally prove this, we have modified the sentence to read "Children generated higher titers of SARS-CoV-2

antibodies than their parents, after being exposed to likely the same viral strain, and antibody titers negatively correlated with age.”

9. Discussion Lines 265-269

The quality of the paediatric humoral immune response is further illustrated by similar binding capacity against SARS-CoV-1 Alpha and beta VOC and a similar neutralization capacity toward Delta VOC compared to adults. This illustrates the generation of a protective long-term humoral immune response also against VOCs after wild type SARS-CoV-2 infection. Difficult to assess unless children have been exposed to VOCs

We agree that a direct assessment of the long-term protection against delta is not possible based upon our data and study design and so have altered the sentence to include “but the quality and duration of this protection can only be estimated following known exposure to VOCs”.

Reviewer #2 (Remarks to the Author):

This is a very well written manuscript that discusses the long term persistence of SARS-CoV-2 specific antibodies and correlation with symptoms. The authors compare pediatric and adult immune responses to SARS-CoV-2 and human CoV. The authors conclude that children had comparable immune response to adults, even with asymptomatic infection. This is an important study as it might help with providing guidance around vaccination.

We thank the reviewer for these supportive comments towards our manuscript.

Major comments:

-patients were not regularly tested by PCR so it is possible that patients were infected from sources outside of their household and at different times during the study (separate from the index case). With that in mind it would be helpful if the authors stated whether or not the children and adults were at school or work during the study period.

We thank the reviewer for this comment. We do not have exact information on whether the participants were at work or not during the study period. National lockdown strategies meant that all child participants were not in school for the majority (approximate 11 ½ months) of the study. While limited PCR capacities and testing requirements mean it is possible that non-index participants could have been exposed to additional sources outside of their household, the relatively low incidence during the first German wave as well as strict lockdown measures, means this can be assumed to be a rare event. We have added the following sentences to the discussion to address this. “Moreover, the limited PCR capacities make it also possible that non-index study participants were additionally exposed to unknown infection sources outside of their household. Given the relatively low incidence in Germany during the first wave (peak incidence spring 2020 45 cases/100.000/7 days) and the strict lockdown measures including school closures, this scenario can be assumed to be a rare event.”

-an additional limitation that should be mentioned is that children may be less likely to accurately report symptoms than adults

Symptoms in younger children were parent-reported. While Dysguesia cannot be reliably assessed in small children, we believe that cough, fever and diarrhea can be reliably described by their parents. Therefore we do not think this is a significant limitation that needs reporting. We have instead added a sentence to the methods stating that symptom information in young children was reported by their parents.

Minor comments:

-Abstract background: suggest clarifying that "long-term persistence of antibodies.... 'are one part' of the immune system that determines..." to clarify that cellular immunity likely also plays an important role

While we obviously fully agree with this comment, we would rather not add this comment into the abstract to not distract for the sake of clarity of the abstract.

-Introduction: line 88: the authors have a minor error with reference 1

We thank the reviewer for spotting this and have correct it.

-Results:

-it would be interesting to know who was the index case in the household (adult versus child) if known and include in results

"The majority of index cases in this study were adults (249 of 272)" has been added to line 411.

-Figure 4: is anything known about the outliers in 4b? Were these individuals more likely to be symptomatic?

We are slightly confused by the term "outliers" in 4b as there are none. We assume the reviewer means 4c? For the seropositive outliers, they are not more likely to be symptomatic.

Discussion:

-minor typo on line 251 should read as "higher titer"

Corrected.

-it is interesting that fever was not a good predictor of SARS-CoV-2 infection in children as respiratory viruses decreased globally during the pandemic due to mitigation measures. It would be interesting for the authors to comment on why they think this might be.

As shown in Supplementary Figure S3, children often seroconverted without any symptoms, not only without fever. While this is a very exciting question, we feel answers would be very speculative at this stage and therefore prefer not to comment in the text.

-Table 1:

was any information collected on co-morbidities for the study participants?

Comorbidities were not systematically evaluated, but we believe they were not very frequent given the age of this cohort

would include # adults and children per household

While this information could potentially be informative, it does not add any further information to what is already listed in the table. Adult would have a median of 2 members, with an IQR of 2-2, Children would have a median of 2 members, with an IQR of 1-2. This is inline with the expectations stated in the methods. Therefore we would prefer to keep the current description of total number of household members.

Reviewer #3 (Remarks to the Author):

This is a well written, strong manuscript containing a timely analysis, in a topic of interest, especially as immunizations are now due to roll out to younger children. The authors report their investigation of 328 households in Germany, where one member of the household had been identified as being SARS CoV-2 positive, with an 11 to 12-months of serologic follow-up (Serum specimens collected roughly 3

months to 11 months after infection of the index case). Uninfected family members served as age matched and environmental controls although did not see much information about controls in the main text, but there are figures with data on seronegative individuals in the supplement. Study cohort was comprised of 548 children and 717 adults from 328 households each with at least one individual with a SARS-CoV-2 RT-PCR proven infection and/or a symptomatic and later serologically proven infection. Participants were recruited during the first wave of the pandemic (May to August 2020) via local health authorities and an in-hospital database of households with at least one laboratory-confirmed SARS-CoV-2 infection.

We thank the reviewer for their positive comments with regards to our manuscript.

Per inclusion criteria it appears that only households with children/ youth between ages 1-18 years were eligible and patients with immunodeficiencies were excluded. This is stated in the methods but figure S2 shows children under age 1 year enrolled so should clarify to be consistent if infants under 1 year of age were enrolled, and if so how many.

We thank the reviewer for drawing our attention to this. No children under 1 were included, this was an error in plotting the graph that it started at 0. The first bar represented children 1-2 years old. We have updated the axis so that it starts at age 1.

The study was conducted during the first wave when no immunizations were available to the population. This would correspond to the time of acute infection. However, immunizations may have been given before collection of specimens in T2 although per Table 1 that was a relatively negligible number of participants (4.2% of adults and 0.3% of children). Only assays measuring antibodies to the nucleocapsid protein would be able to distinguish antibodies from vaccine to antibodies generated by natural infection. Nucleocapsid antibodies would not be detected by the EuroImmuno Anti-SARS-CoV-2 ELISA and the Siemens Healthineers SARS-CoV-2 IgG (sCOVG) which measure antibodies to spike protein. The authors mention that in T2 the assay used for all specimens was the Roche Elecsys Electrochemiluminescence immunoassay (ECLIA) which measures antibodies to the nucleocapsid of SARS-CoV-2 and the MULTICOV-AB™1, a bead-based 136 multiplex immunoassay that simultaneously analyzes 23 antigens from SARS-CoV-2. This information is included in the supplement but should be highlighted in the main manuscript too as it is an important point to make regarding interpretation of results.

We thank the reviewer for this comment regarding immunisations. We have now transferred all supplementary methods to the main manuscript file as per the journal's requests.

Pseudovirus neutralization assays were performed in a subset of 385 specimens, so it is important to include in the study limitation section in the discussion that the neutralization results do not apply to all evaluable serum specimens. The statements in the discussion about neutralization results are very broad and there is no mention there that only a subset of specimens were analyzed.

We have included "a subset of" to line 232 of the results, as well as a sentence in the limitations paragraph, stating again that only a subset were analysed and that caution should be used in applying it to all samples. "It should be also stated that while all samples were analysed using a range of serological assays, only a subset of samples were analysed for their neutralizing capabilities, and as such, caution should be applied in extrapolating the implications regarding the neutralizing response to all study participants."

Overall the results are very interesting and substantiated by other reports in which children were less frequently seropositive (33% and 37% respectively at the 2 timepoints) as compared to adults (58% and 40% respectively at the 2 timepoints). Most participants were asymptomatic or mildly symptomatic with a very low hospitalization rate. Symptoms were associated with increasing age. It is important to remember that this was a cohort from early on in the pandemic, before variants of concern such as

Delta strains emerged. Children sustained longer duration of antibody responses than adults in this study, with seroreversion being more frequently seen in adults than children as stated in the results section: “only 3·8% of children, but 17·1% of adults seroreverted between T1 and T2 (Table 1).” However, figure S6 in the supplement section states that seroreversion occurs at the same rate in adults and children, so this is confusing to the reader. The authors should clarify if there are any discrepancies in the text.

We thank the reviewer for these comments. There was a difference between seroreversion (testing negative on at least 2 assays) versus the antibody decay in S6 Figure (ratio of signal remaining). We have remained S6 Figure antibody decay instead of seroreversion to hopefully avoid confusion.

Also interesting results was that there was no difference in antibody responses between asymptomatic and symptomatic infections in this cohort. This could be due to the fact that the cohort did not have many participants with severe illness, so the absence of a notable difference could be due to the fact that people were not severely ill in general, and not because symptomatic individuals produce the same amount of antibody that asymptomatic individuals. This should be mentioned in the discussion, otherwise interpretation of results could be misleading. Many studies have shown that individuals with severe illness tend to develop higher titer antibodies over time which the authors do mention in the discussion.

We thank the reviewer for these comments. We think they reporting of no differences in responses between asymptomatic and symptomatic infections was already reported in the discussion (lines 295 to 303). We have added an additional clarifier “as in our cohort” to line 299 to help.

Interest finding that there was no association between antibody responses to HCoV and SARS-CoV-2 antibody responses in children and adults, which indicates that there is no cross-protection conferred by antibodies to other coronaviruses. This had been postulated as a reason why children are not generally so ill when they acquire SARS- CoV-2. This is an important finding in this study.

We are grateful to the reviewer for appreciating the significance of this finding.

There have been other household transmission studies evaluating pediatric and adult populations with SARS CoV-2 infection, so not clear whether the authors’ claim that this is the largest multi-centric study conducted to date in pediatric populations still holds. Some of the other studies were from the early phase of the pandemic such as that of Lugon P et al. SARS-CoV-2 Infection Dynamics in Children and Household Contacts in a Slum in Rio de Janeiro. Pediatrics. 2021. Others were from later pandemic stages with circulation of more contagious VOC such as Liu P et al Pediatric Household Transmission of SARS CoV-2 Infection- Los Angeles County, December 2020 to February 2021. Pediatric Infectious Diseases Journal, October 2021; Dawood FS et al. Incidence rates, household infection risk, and clinical characteristics of SARS CoV-2 infection among children and adults in Utah an New York City, NY. JAMA Pediatrics 2021, among others. It would be of interest if the authors included a paragraph in their discussion comparing their findings with that of the similar cohort studies, such as the ones mentioned above.

While we agree with the reviewer that there are other household transmission studies of similar size, most of these studies use a single timepoint for assessing serological status, and none of the studies – to our understanding – performed this multiparametric serological analysis in a prospective manner over the course of a year following infection. Thus we would be grateful if we could leave this first sentence unchanged.

Some minor comments:

Introduction, line 88: Would just end sentence with reference rather than state reviewed in 1.

As per reviewer 1, this has already been changed.

Discussion, line 260: Would rewrite to: with children seroreverting at a significantly slower pace than adults. Discussion, line 261: Would change to: Children generated higher titers...

Modified as suggested.

Discussion, line 262: Would change to: antibody titers negatively correlated with age.

Changed as suggested.